# Performance Evaluation of Stewart-Gough Flight Simulator Based on $\mathcal{L}_1$ Adaptive Control

**Jiangwei Zhao, Dongsu Wu \* and Hongbin Gu**

Advanced Flight Simulation Laboratory, Nanjing University of Aeronautics and Astronautics, Nanjing 211100, China; jiangwei_zhao@nuaa.edu.cn (J.Z.); ghb@nuaa.edu.cn (H.G.)
\* Correspondence: tissle@nuaa.edu.cn

**Abstract:** In the design of the six degrees of freedom (6-DOF) flight simulation system, the unmodeled dynamic, transient performance and steady-state performance of the system are generally concerned. Considering that the model of flight simulation system is highly nonlinear and requires high response speed and high stability, this paper applies $\mathcal{L}_1$ adaptive controller to the control of flight simulation platform. The controller has a low-pass filter in feedback loop to avoid high frequencies in the control signals, and the required transient performance can be enhanced by increasing the adaptive gain, which can improve the transient, stability, and smoothness of the flight simulator platform. The performance of the $\mathcal{L}_1$ adaptive controller is obtained by comparison with the traditional model reference adaptive controller (MRAC). In addition to maintaining the good transient response of MRAC, the $\mathcal{L}_1$ adaptive controller improves the stability of the system. The output amplitude of the actuator is reduced by 39.95%, which effectively reduces the performance requirements of the actuator. Some additional experimental evaluations are carried out to show the performance of the controller.

**Keywords:** $\mathcal{L}_1$ adaptive control; Stewart; Gough; flight simulator; MRAC

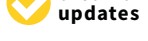



## 1. Introduction

According to Boeing's annual statistics on flight accidents, LOSS-OF-CNOTROL In-Flight (LOC-I) has become the primary cause of fatal commercial aviation fatalities [1–4], and there is no effective way to avoid the aircraft entering all abnormal conditions. Therefore, the pilot should adequately train by practicing the recovery technique many times on flight simulators before maneuvers on a real aircraft. However, for safety reasons, the training of upset recovery can only be carried out on aerobatics or flight simulators [5]. Thus, the flight simulator becomes the only tool that can safely, cost-effectively, and systematically carry out such abnormal state correction training [6]. Abnormal states such as high-altitude stall, wind shear, and turbulence are usually accompanied by the abnormal attitude of the aircraft and the large changes of angular velocity and acceleration, which involve higher requirements of the transient characteristics and anti-high frequency interference ability of the flight simulator control system. The six degrees of freedom (6-DOF) Stewart-Gough platform is used in the design of flight simulators because of its advantages of multi-DOF, high precision, and high load. Research on control strategy of the platform is the key to improve the motion fidelity of simulator.

In the last decades, there have been some remarkable attempts to control the Stewart-Gough flight simulator platform in the task space [7–10]. Two classes of approaches are being actively studied to maintain the performance of the platform in the presence of parameter uncertainties: Robust control [11–14] and adaptive control [15–18]. The model reference adaptive controller (MRAC) is a typical structure of the adaptive control system. One advantage of the MRAC method is that the accuracy of the platform will be improved over time, because the MRAC mechanism constantly extracts parameter information from tracking errors. However, when the adaptive rate is small, a large amplitude of control

signal is easy to appear during the transient. When the adaptive law is large, the system will have high-frequency oscillation. The improvement of the transient performance of adaptive controllers has been addressed from various perspectives in numerous efforts [19–26]. Among these efforts, the achievement of $\mathcal{L}_1$ adaptive control proposed in literature [26] is the most outstanding, and the $\mathcal{L}_1$ adaptive controller is applied to the control field of flight simulation platform. Due to the advantage of the separation of control law and adaptive law, the low-pass filter is added to the control input to avoid the high-frequency noise in the control signal. The structure diagram of $\mathcal{L}_1$ adaptive flight simulation control system is shown in Figure 1.

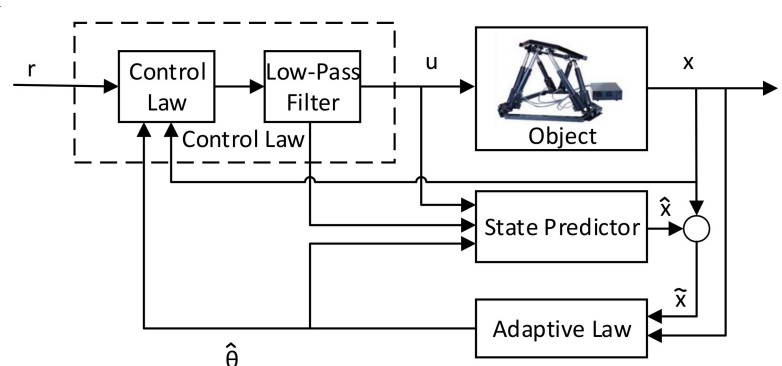

**Figure 1.** Structure diagram of the $\mathcal{L}_1$ adaptive flight simulation control system.

An $\mathcal{L}_1$ adaptive controller for Stewart-Gough platform is presented in this paper. In this method, $\mathcal{L}_1$ gain is introduced to ensure the asymptotic stability of the system. Another feature is the introduction of the low-pass filter in the design of control rate to avoid high frequency in the control signal, and the filter can also be used to reduce the output amplitude of the actuator in transient state. This paper is organized as follows. Section 1 states the background of $\mathcal{L}_1$ adaptive control development. Section 2 gives the dynamic model of the Stewart-Gough flight simulation platform. In Section 3, the $\mathcal{L}_1$ adaptive controller based on 6-DOF Stewart-Gough flight simulator is designed, and its simulation results analysis is in Section 4. Finally, in Section 5, the paper is summarized.

## 2. Dynamic Model of Stewart-Gough Flight Simulation Platform

In this paper, the dynamic model of Stewart-Gough flight simulation platform is built based on the virtual working principle. The experimental platform is shown in Figure 2. The virtual work equation, according to the authors of [27], is:

$$\delta q^T \tau + \delta x_p^T \hat{F}_p + \sum_i^6 \left( \delta x_{i_1}^T \hat{F}_{i_1} + \delta x_{i_2}^T \hat{F}_{i_2} \right) = 0, \tag{1}$$

where $\delta q = [\delta q_1 \dots \delta q_6]^T$ denotes the virtual displacement vector of actuated joints, $\tau = [d\tau_1 \dots d\tau_6]^T$ denotes the force and torque vector acting on the actuated joints, and $\delta x_p = \left[ \delta x_p, \delta y_p, \delta z_p, \delta \theta_{x_p}, \delta \theta_{y_p}, \delta \theta_{z_p} \right]^T$ denotes the virtual variables of a contacting point of the moving platform. Furthermore, $\delta x_{i_1} = \left[ \delta x_{i_1}, \delta y_{i_1}, \delta z_{i_1}, \delta \theta_{x_{i_1}}, \delta \theta_{y_{i_1}}, \delta \theta_{z_{i_1}} \right]^T$ represents the virtual variables vector of the upper legs center of gravity (c.o.g.). Similarly, $\delta x_{i_2}$ denotes the virtual displacement vector c.o.g. of down legs, $\hat{F}_p$ denotes the inertial forces at the c.o.g. of the moving platform, and $\hat{F}_{i_1}$ and $\hat{F}_{i_2}$ are the inertial forces at the c.o.g. of up legs and at the c.o.g. of the down legs, respectively, which are defined from the relationship, according the authors of [27], as:

$$\hat{F}_p = \begin{bmatrix} f_d + m_p g - m_p \ddot{x}_p \\ n_d - I_p \dot{\omega}_p - \omega_p \times I_p \omega_p \end{bmatrix} = \tau_{d,p} - M_p \ddot{x}_p - C_p \dot{x}_p - G_p, \tag{2}$$

$$\hat{F}_{i_1} = \begin{bmatrix} m_{i_1}g - m_{i_1}\ddot{x}_{i_1} \\ -I_{i_1}\dot{\omega}_{i_1} - \omega_{i_1} \times I_{i_1}\omega_{i_1} \end{bmatrix} = -M_{i_1}\ddot{x}_{i_1} - C_{i_1}\dot{x}_{i_1} - G_{i_1}, \tag{3}$$

$$\hat{F}_{i_2} = \begin{bmatrix} m_{i_2}g - m_{i_2}\ddot{x}_{i_2} \\ -I_{i_2}\dot{\omega}_{i_2} - \omega_{i_2} \times I_{i_2}\omega_{i_2} \end{bmatrix} = -M_{i_2}\ddot{x}_{i_2} - C_{i_2}\dot{x}_{i_2} - G_{i_2}, \tag{4}$$

where $M_p$, $M_{i_1}$, and $M_{i_2}$ denote the mass matrix of the moving platform, of the up legs, and of the down legs given in Equation (5); and $C_p$, $C_{i_1}$, and $C_{i_2}$ denote the Coriolis and centrifugal matrix of the moving platform, of the up legs, and of the down legs given in Equation (6). Furthermore, $G_p$, $G_{i_1}$, and $G_{i_2}$ denote the gravity vector of the moving platform, of the up legs, and of the down legs given in Equation (7).

$$M_p = \begin{bmatrix} m_p1 & 0 \\ 0 & I_p \end{bmatrix} M_{i_1} = \begin{bmatrix} m_{i_1}1 & 0 \\ 0 & I_{i_1} \end{bmatrix} M_{i_2} = \begin{bmatrix} m_{i_2}1 & 0 \\ 0 & I_{i_2} \end{bmatrix}, \tag{5}$$

$$C_{i_1} = \begin{bmatrix} 0 & 0 \\ 0 & \omega_{i_1} \times I_{i_1} \end{bmatrix} C_{i_2} = \begin{bmatrix} 0 & 0 \\ 0 & \omega_{i_1} \times I_{i_1} \end{bmatrix} C_p = \begin{bmatrix} 0 & 0 \\ 0 & \omega_p \times I_p \end{bmatrix}, \tag{6}$$

$$G_p = \begin{bmatrix} -m_pg \\ 0 \end{bmatrix} G_{i_1} = \begin{bmatrix} -m_{i_1}g \\ 0 \end{bmatrix} G_{i_2} = \begin{bmatrix} -m_{i_2}g \\ 0 \end{bmatrix}, \tag{7}$$

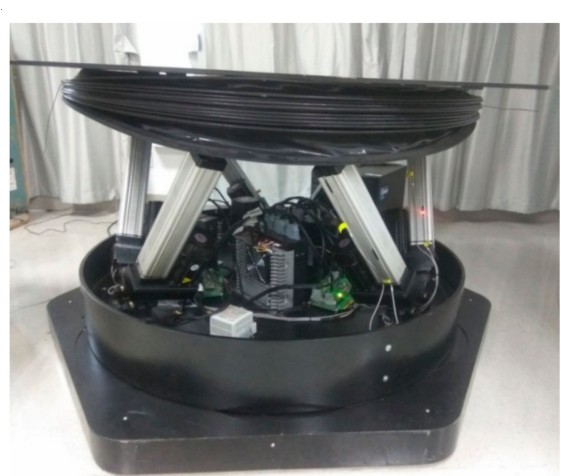

**Figure 2.** Six degrees of freedom (6-DOF) flight simulation experiment platform.

The virtual displacement $\delta q$ of the actuated joints and the virtual displacement of the center of gravity $\delta x_{i_1}$ and $\delta x_{i_2}$ of the legs, however, must be compatible with the kinematic constraints imposed by the virtual displacement of the moving platform. Therefore, it is necessary to obtain the Jacobian matrix $J_p$ and $J_{i_1}$, $J_{i_2}$ of the virtual displacement from the center of gravity of the actuated joints and the legs to the virtual displacement of the moving platform. For the Stewart-Gough platform, according to Tsai LW in [5], we have:

$$\delta q = J_p \delta x_p, \tag{8}$$

$$\delta x_{i_1} = J_{i_1} \delta x_p, \tag{9}$$

$$\delta x_{i_2} = J_{i_2} \delta x_p, \tag{10}$$

Substituting Equations (8)–(10) into (1) yields:

$$J_p^T \tau + \hat{F}_p + \sum_i^6 \left( J_{i_1}^T \hat{F}_{i_1} + J_{i_2}^T \hat{F}_{i_2} \right) = 0, \tag{11}$$

Substituting Equations (5)–(7) into (11) and simplifying yields the general closed-form dynamic formulation of Stewart-Gough platform represented by Equation (12).

$$M\ddot{X} + C\dot{X} + G = F, \tag{12}$$

where $M$ denotes the mass matrix of the Stewart-Gough platform given in Equation (13), $C$ denotes the Coriolis and centrifugal matrix of the Stewart-Gough platform given in Equation (14), $G$ denotes the gravity vector of the Stewart-Gough platform given in Equation (15), $F$ denotes the generalized actuated force and external disturbance force of the Stewart-Gough platform given in Equation (16), and $X$ denotes the vector of moving platform motion variables $X = \left[ x_p, y_p, z_p, \theta_{x_p}, \theta_{y_p}, \theta_{z_p} \right]^T$. $\dot{X}$ denotes the moving platform twist vector, and $\ddot{X}$ denotes the moving platform acceleration vector.

$$M = \left( M_p + \sum_i^6 \left( J_{i_1}^T M_{i_1} J_{i_1} + J_{i_2}^T M_{i_2} J_{i_2} \right) \right), \tag{13}$$

$$C = \left( C_p + \sum_i^6 \left( J_{i_1}^T C_{i_1} J_{i_1} + J_{i_2}^T C_{i_2} J_{i_2} + J_{i_1}^T M_{i_1} J_{i_1} + J_{i_2}^T M_{i_2} J_{i_2} \right) \right), \tag{14}$$

$$G = \left( G_p + \sum_i^6 \left( J_{i_1}^T G_{i_1} + J_{i_2}^T G_{i_2} \right) \right), \tag{15}$$

$$F = J_p^T \tau + \tau_{d,p}, \tag{16}$$

## 3. Controller Design

### 3.1. The Architecture of the MRAC Adaptive Controller

$\mathcal{L}_1$ adaptive control is developed on the basis of MRAC. Therefore, we consider the architecture of the 6-DOF Stewart-Gough flight simulation platform based on MRAC given by Figure 3 [28].

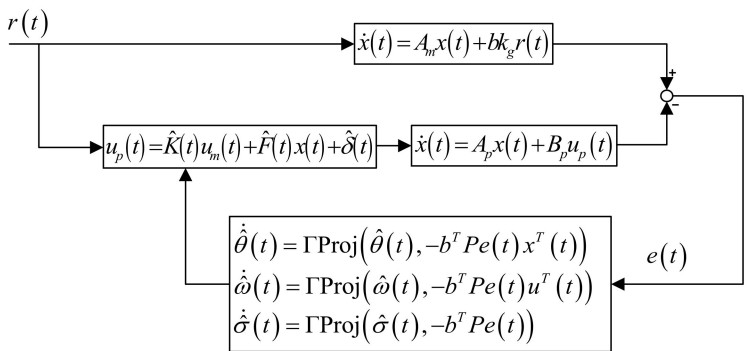

**Figure 3.** Closed-loop model reference adaptive controller (MRAC) architecture.

The state equation and output equation of the 6-DOF Stewart-Gough flight simulation platform is:

$$\dot{x}(t) = A_p x(t) + B_p u_p(t)$$
$$y(t) = c^T x(t) \tag{17}$$

where $x(t) \in \mathbb{R}^m$ denotes the state of the system, $A_p \in \mathbb{R}^{m \times m}$ denotes the state matrix, $B_p \in \mathbb{R}^{m \times n}$ denotes the input matrix, $u_p(t) \in \mathbb{R}^n$ denotes the control input, $c^T \in \mathbb{R}^{m \times m}$ denotes the output matrix, and $y(t) \in \mathbb{R}^n$ denotes the output of the system.

The reference system of the Stewart-Gough flight simulation platform, given by [29]:

$$\dot{x}(t) = A_m x(t) + b k_g u_m(t)$$
$$y(t) = c^T x(t) \tag{18}$$

where $A_m \in R^{m \times m}$ denotes a Hurwitz matrix, $b \in R^m$ denotes a known constant vector, $(A_m, b)$ is controllable, and $k_g = \lim\limits_{s \to 0} \left(1/\left(c^T(sI - A_m)^{-1}b\right)\right)$ denotes a known constant, which can ensure that the system has zero steady-state error when the input is constant.

The model reference adaptive controller is given by:

$$u_p(t) = \hat{K}(t)u_m(t) + \hat{F}(t)x(t) + \hat{\delta}(t) \tag{19}$$

where $\hat{K}(t) \in \mathbb{R}^{m \times m}$ denotes the feed-forward gain matrix of the system, $\hat{F}(t) \in \mathbb{R}^{m \times n}$ denotes the feedback compensation matrix of the system, and $\hat{\delta}(t)$ denotes the input disturbances.

### 3.2. The Architecture of the $\mathcal{L}_1$ Adaptive Controller

In order to ensure that the input and output of uncertain linear systems can track the input and output of the reference system in real time, we can reform the MRAC of the Stewart-Gough flight simulation platform given in Equation (18) into the following control structure:

$$\dot{x}(t) = \left(A_p - B_p F^T\right)x(t) + B_p K(t)u(t) + B_p\left(F^T - F(t)\right)x(t) + B_p\hat{\delta}(t)$$

$$y(t) = c^T x(t) \tag{20}$$

where the system is divided into a linear time invariant (LTI) and linear time varying (LTV) system. After simplification, the system yields [30]:

$$\dot{x}(t) = A_m x(t) + b(\omega(t)u(t) + \theta(t)x(t) + \sigma(t))$$

$$y(t) = c^T x(t) \tag{21}$$

where $x(t) \in \mathbb{R}^m$ represents the state of the system, $u(t) \in \mathbb{R}^n$ represents the control input, $c^T \in \mathbb{R}^{n \times m}$ represents the output matrix, $b \in \mathbb{R}^{m \times n}$ represents the known constant matrix, $y(t) \in \mathbb{R}^n$ denotes the regulated output, $\omega(t) \in \mathbb{R}^{n \times n}$ denotes the input uncertainly parameter of the model, $\theta(t) \in \mathbb{R}^{n \times m}$ is the uncertainly parameter of the model itself, and $\sigma(t) \in \mathbb{R}^{n \times 1}$ represents the unmatched disturbances. $A_m \in \mathbb{R}^{m \times m}$ represents a Hurwitz matrix, and $(A_m, b)$ is controllable and $A_m - A = b\theta(t)$.

For the linear time invariant parameterized system in Equation (21), we give the following state observer model:

$$\dot{\hat{x}}(t) = A_m\hat{x}(t) + b\left(\hat{\omega}(t)u(t) + \hat{\theta}(t)x(t) + \hat{\sigma}(t)\right)$$

$$\hat{y}(t) = c^T \hat{x}(t) \tag{22}$$

where $\hat{\omega}(t)$, $\hat{\theta}(t)$ and $\hat{\sigma}(t)$ denote the parameter estimates of $\omega(t), \theta(t)$ and $\sigma(t)$. The block diagram of the closed-loop system is shown in Figure 4.

Next, we use Lyapunov stability theory to obtain adaptive law, considering the following Lyapunov function candidate [30]:

$$V = e^T(t)Pe(t) + \frac{1}{\Gamma}\left(\widetilde{\omega}^T(t)\widetilde{\omega}(t) + \widetilde{\theta}^T(t)\widetilde{\theta}(t) + \widetilde{\sigma}^T(t)\widetilde{\sigma}(t)\right) \tag{23}$$

where $e(t) = \hat{x}(t) - x(t)$, $\widetilde{\omega} = \hat{\omega}(t) - \omega(t)$, $\widetilde{\theta} = \hat{\theta}(t) - \theta(t)$, $\widetilde{\sigma} = \hat{\sigma}(t) - \sigma(t)$ denotes the tracking errors between state predictor and system, $\Gamma > 0$ denotes the adaptation gain, and $P = P^T > 0$ denotes the solution of equation $A_m^T P + PA_m = -Q, Q > 0$. The adaptive laws can be obtained by solving algebraic Lyapunov equation. Using projection operator [16] $Proj(\cdot, \cdot)$ to prevent parameter drift, we have:

$$\dot{\hat{\theta}}(t) = \Gamma Proj\left(\hat{\theta}(t), -b^T Pe(t)x^T(t)\right)$$

$$\dot{\hat{\omega}}(t) = \Gamma Proj\left(\hat{\omega}(t), -b^T Pe(t)u^T(t)\right)$$

$$\dot{\hat{\sigma}}(t) = \Gamma Proj\left(\hat{\sigma}(t), -b^T Pe(t)\right) \tag{24}$$

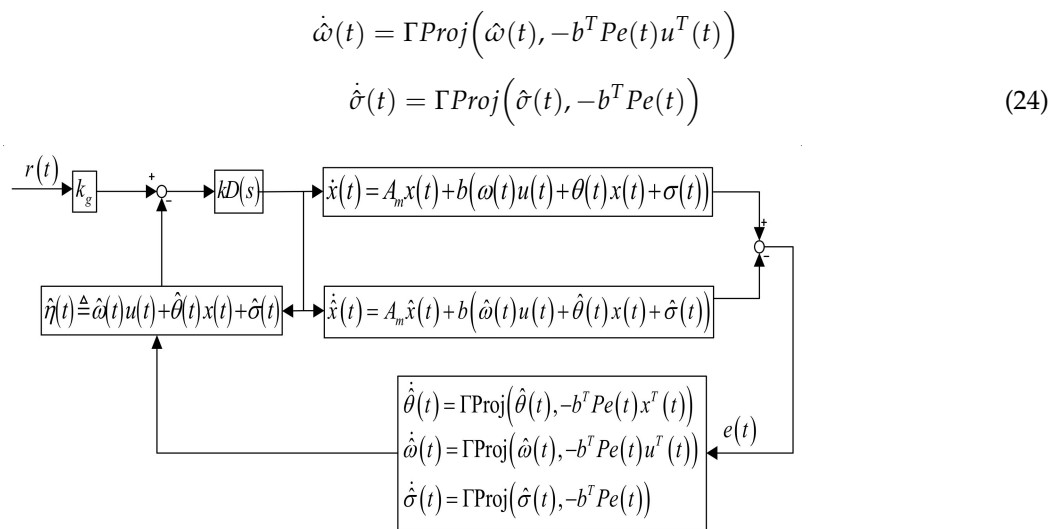

**Figure 4.** Closed-loop $\mathcal{L}_1$ adaptive controller architecture.

The fundamental difference between the MRAC in Figure 3 and the predictor-based of $\mathcal{L}_1$ adaption in Figure 4 is the design of the control law. The design of error signal and controller are not independent in MRAC system, but they are independent in $\mathcal{L}_1$ adaptive control system. Therefore, in order to ensure the stability of the system, besides using the Lyapunvo theory to design the stability of the error signal, the stability design of the control law is also needed. The control law is design through gain feedback, given by [31]:

$$u(s) = -kD(s)\left(\hat{\eta}(s) - k_g r(s)\right) \tag{25}$$

where $r(s)$ and $\hat{\eta}(s)$ denote the Laplace transforms of $r(t)$ and $\hat{\eta}(t) \triangleq \hat{\omega}(t)u(t) + \hat{\theta}(t)x(t) + \hat{\sigma}(t)$, $k_g \triangleq -1/\left(c^T A_m^{-1}b\right)$ denotes the constant matrix, $k > 0$ denotes the feedback gain, and $D(s)$ denotes the strictly proper transfer function, leading to a strictly proper stable [32]:

$$D(s) = \frac{\omega KD(s)}{1 + \omega KD(s)} \tag{26}$$

In which

$$L \triangleq \max_{\theta \in \Theta} \| \theta \|_{\mathcal{L}1} \triangleq \max_i \left(\sum_j |\theta_{ij}|\right)$$

$$H(s) \triangleq (sI - A_m)^{-1}b$$

$$G(s) \triangleq H(s)(1 - C(s)) \tag{27}$$

The controller needs to satisfy the following $\mathcal{L}_1$-norm conditions for the stability of the system [33]:

$$\| G(s) \|_{\mathcal{L}1} L < 1 \tag{28}$$

## 4. Experiment Verification

### 4.1. Plant

The modeling parameters required for the experiment are from the 6-DOF Stewart-Gough flight simulator platform shown in Figure 2, and the dynamic model and the $\mathcal{L}_1$ adaptive controller were implemented in MatLab. The platform was composed of six cylinders, six universal hinges, and two platforms. The lower platform was fixed on the foundation plane. With the help of the telescopic motion of six cylinders, the motion of the upper platform with six degrees of freedom (X,Y,Z,$\alpha$,$\beta$,$\gamma$) in space can be completed, so that all kinds of space motion posture can be simulated. The upper end points of six cylinders formed three fulcrums of the upper platform, and the lower end points of six cylinders

formed three fulcrums of the lower platform. The upper and lower three fulcrums were, respectively, on the assumed circle, and were divided equally by 120 degrees, which were the vertices of two equilateral triangles. The geometric and inertial parameters of the platform are shown in Table 1.

**Table 1.** Platform geometric and inertial parameters.

| Variable | Value | Units |
|----------|-------|-------|
| $radius_{lower}$ | 0.504 | m |
| $radius_{upper}$ | 0.504 | m |
| $\theta_{A_i}$ | [30,30,150,150,270,270] | deg |
| $\theta_{B_i}$ | [0,60,120,180,240,300] | deg |
| $m_p$ | 28.7 | kg |
| $m_{i_1}$ | 2.77 | kg |
| $m_{i_2}$ | 0.54 | kg |
| $c_{i_1}$ | 0.5456 | m |
| $c_{i_2}$ | 0.305 | m |
| $I_{xx,p}$ | 1.13 | kg m$^2$ |
| $I_{yy,p}$ | 1.13 | kg m$^2$ |
| $I_{zz,p}$ | 2.23 | kg m$^2$ |
| $I_{xx,c_{i_1}}$ | 0.21 | kg m$^2$ |
| $I_{yy,c_{i_1}}$ | 0.21 | kg m$^2$ |
| $I_{zz,c_{i_1}}$ | 0.001 | kg m$^2$ |
| $I_{xx,c_{i_2}}$ | 0.0677 | kg m$^2$ |
| $I_{yy,c_{i_2}}$ | 0.0677 | kg m$^2$ |
| $I_{zz,c_{i_2}}$ | 0.000114 | kg m$^2$ |
| $g$ | [0,9.8,0] | N/kg |

### 4.2. Path Planning

We built the dynamic model of the flight simulation platform according to Equation (12), and the controller calculated the required actuator force or torque to make the robot follow the desired translation and orientation trajectory. We used the translation and orientation variable as the coordinate of the platform defined by $X = [x \ \theta]^T$, in which the translation is represented by $x = \begin{bmatrix} x_p \ y_p \ z_p \end{bmatrix}^T$, while the moving platform orientation is represented by $\theta = \begin{bmatrix} \theta_{x_p} \ \theta_{y_p} \ \theta_{z_p} \end{bmatrix}^T$. Both of the translation and orientation planning were built using a sinusoidal motion given by:

$$r = \begin{bmatrix} x_{plan} \\ \theta_{plan} \end{bmatrix} = \begin{bmatrix} x_r + A_r sin\left(\omega_{plan}t\right) \\ A_o sin\left(\omega_{plan}t\right) \end{bmatrix} \tag{29}$$

We used the twist velocity variable as the coordinate of the platform defined by $\dot{r} = \begin{bmatrix} v_{plan} \omega_{plan} \end{bmatrix}^T$, and the twist velocity planning was obtained after differentiating the Equation (30) given by:

$$\dot{r} = \begin{bmatrix} v_{plan} \\ \omega_{plan} \end{bmatrix} = \begin{bmatrix} A_r \omega_{plan} cos\left(\omega_{plan}t\right) \\ A_o \omega_{plan} cos\left(\omega_{plan}t\right) \end{bmatrix} \tag{30}$$

The values used in path planning of the 6-DOF Stewart-Gough flight simulation platform simulations are given in Table 2.

**Table 2.** Platform geometric and inertial parameters.

| Variable | Value | Units |
|:---:|:---:|:---:|
| $x_0$ | $[0, 0, 0.48]^T$ | m |
| $\dot{x}_0$ | $[0, 1.0, 0]^T$ | m$^2$ |
| $r_{step}$ | $[0.2, 0.3, 0.78, 0.25, 0.5, 1.0]^T$ | m |
| $\dot{r}_{step}$ | $[0, 0, 0, 0, 0, 0]^T$ | m$^2$ |
| $x_r$ | $[0, 0, 0.2]^T$ | m |
| $A_r$ | $[0.2, 0.3, 0.14]^T$ | m |
| $A_o$ | $[0.25, 0.5, 1.0]^T$ | deg |
| $\omega_{plan}$ | 1.6 | rad/s |
| $k$ | 80 | - |
| $L$ | 150,000 | - |
| $\omega_n$ | diag[10,10,10,10,10,10] | - |
| $\xi$ | diag[0.7,0.7,0.7,0.7,0.7,0.7] | - |
| $\omega$ | $\omega \in [0.8, 1.2]$ | - |
| $\theta(t)$ | $|\theta(t)| \leq 3800$ | - |
| $\sigma(t)$ | $|\sigma(t)| \leq 400$ | - |

*4.3. Control System*

In the initial position $x_0 = [0\ 0\ 0.48]^T$ and $\dot{x}_0 = [0\ 1.0\ 0]^T$ of the flight simulation platform, we calculated the dynamics equation parameters of the system in Equation (12) according to the geometrical and inertial parameters in Table 1.

$$A = \begin{bmatrix} 0_{6\times6} & I_{6\times6} \\ 0_{6\times6} & -M_0^{-1}C_0 \end{bmatrix} \qquad B = \begin{bmatrix} 0_{6\times6} \\ M_0^{-1} \end{bmatrix}$$

$$M_0 = \begin{bmatrix} 39.1978 & -0.0000 & 0.0000 & 0.0000 & 0.2242 & -0.0000 \\ -0.0000 & 39.1978 & -0.0000 & -0.2242 & 0.0000 & -0.0000 \\ 0.0000 & -0.0000 & 33.8082 & 0.0000 & 0.0000 & -0.0000 \\ -0.0000 & -0.2242 & 0.0000 & 1.7788 & -0.0000 & -0.0000 \\ 0.2242 & 0.0000 & -0.0000 & -0.0000 & 1.7788 & 0.0000 \\ -0.0000 & -0.0000 & -0.0000 & -0.0000 & 0.0000 & 4.6912 \end{bmatrix}$$

$$C_0 = \begin{bmatrix} 0.0000 & 0.6085 & 0.0000 & 0.6912 & -0.0000 & 0.5702 \\ 0.6085 & -0.0000 & -11.4379 & 0.0000 & -0.6912 & -0.0000 \\ 0.0000 & 4.5986 & 0.0000 & 0.2422 & -0.0000 & 0.0000 \\ 0.6909 & 0.0000 & -0.2422 & -0.0000 & -0.0610 & 0.0000 \\ 0.0000 & -0.6909 & -0.0000 & 0.0610 & 0.0000 & -0.0000 \\ -0.0000 & -0.0000 & 0.0000 & -0.0000 & 1.1043 & -0.0000 \end{bmatrix}$$

$$G_0 = \begin{bmatrix} 0 & 0 & 329.5711 & 0 & 0 & 0 \end{bmatrix}^T \tag{31}$$

where $M_0$, $C_0$, and $G_0$ denote the initial date of $M$, $C$, and $G$.

According to the 6-DOF Stewart-Gough flight simulation platform dynamic equation shown in Equation (12), we chose the following differential equation as the reference model:

$$\ddot{X} + 2\xi\omega_n\dot{X} + \omega_n^2 = \omega_n^2 \tag{32}$$

where $\xi \in R$ denotes the damping ratio and $\omega_n$ denotes the undamped natural frequency.

The state equation of the reference model is:

$$\dot{x}(t) = A_m x(t) + bk_g u_m(t) \tag{33}$$

where

$$A_m = \begin{bmatrix} 0_{6\times6} & I_{6\times6} \\ -\omega_n^2 & -2\xi\omega_n \end{bmatrix}$$

$$b = \begin{bmatrix} 0_{6\times6} & M_0^{-1} \end{bmatrix}^T$$

$$k_g = -\left(cA_m^{-1}b\right)^{-1} \tag{34}$$

Further, we can derive the initial value of $\omega(0)$, $\theta(0)$, and $\sigma(0)$, where

$$\hat{\omega}(0) = [I_{6\times6}]$$

$$\hat{\theta}(0) = \begin{bmatrix} M_0\omega_n^2 & 2M_0\xi\omega_n - C_0 \end{bmatrix}$$

$$\hat{\sigma}(0) = [-G_0] \tag{35}$$

The $\mathcal{L}_1$ adaptive controller was designed based on Equations (22), (24), and (25) as follows:

$$\dot{\hat{x}}(t) = A_m\hat{x}(t) + b\big(\hat{\omega}(t)u(t) + \hat{\theta}(t)x(t) + \hat{\sigma}(t)\big)$$

$$\dot{\hat{\omega}}_j(t) = \Gamma Proj\Big(\hat{\omega}_j(t), -b^T Pe(t)u^T(t)\Big)_{\{j,j\}}$$

$$\dot{\hat{\theta}}_i(t) = \Gamma Proj\Big(\hat{\theta}_i(t), -b^T Pe(t)x^T(t)\Big)_{\{i,:\}}$$

$$\dot{\hat{\sigma}}(t) = \Gamma Proj\Big(\hat{\sigma}(t), -b^T Pe(t)\Big) \tag{36}$$

where $[\ ]_{\{j,j\}}$ $j = [1, \ldots, 6]$ denotes the $j^{th}$ row $j^{th}$ column element of the matrix $[\ ]$, and $[\ ]_{\{i,:\}}$ $i = [1, \ldots, 12]$ denotes the $i^{th}$ row vector of the matrix $[\ ]$,. The control signal is given by:

$$D(s) = \frac{\omega KD(s)}{1 + \omega KD(s)} \tag{37}$$

where $D(s) = 1/s$, $K = kI_{6\times6}$, $k > 0$.

$$\hat{\eta}(t) \triangleq \hat{\omega}(t)u(t) + \hat{\theta}(t)x(t) + \hat{\sigma}(t)$$

It can be derived straightforwardly that $L = 150,000$, while $\parallel G(s) \parallel_{\mathcal{L}1} L$ can be obtained. Figure 5 shows $\lambda = \parallel G(s) \parallel_{\mathcal{L}1} L$ with respect to the bandwidth of the low-pass filter $\omega$. Let $\omega = 0.8$, obtain $k = 80$ that satisfies the $\mathcal{L}_1$-gain upper bound. Table 2 has the constant values used in the $\mathcal{L}_1$ adaptive controller.

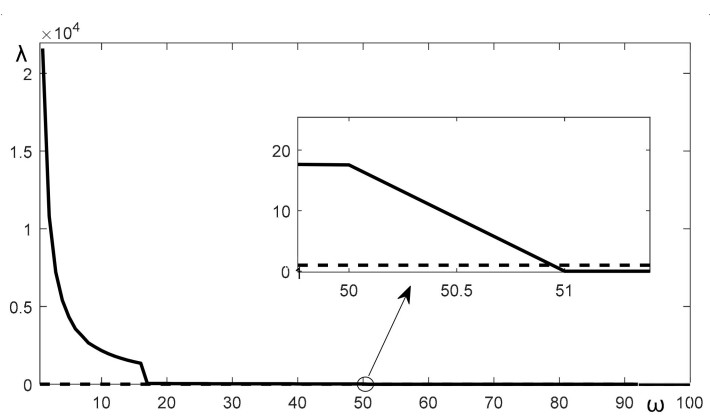

**Figure 5.** $\lambda$ with respect to $\omega$.

In the design of the $\mathcal{L}_1$ controller, we needed to estimate the boundary of parameters. Here, we mainly used the experimental method to estimate the boundary of $\omega$, $\theta$ and $\sigma$. Take the parameter boundary estimation of $\sigma$ as an example. First, set a larger boundary, then make the platform control input $r = \big[0.2sin(\pi t), 0.3sin(\pi t), 0.48 + 0.14sin(\pi t), \frac{\pi}{12}sin(\pi t), \frac{\pi}{6}sin(\pi t), \frac{\pi}{3}sin(\pi t)\big]$ and impose external disturbance on the moving platform to obtain the accurate boundary of parameter $\sigma$ under the complex state of the moving platform. The curve data of parameter $\sigma$ is

shown in Figure 6, and the bounds of $\sigma$ is $|\sigma_i(t)| \leq 400$, $i = [1, \ldots, 6]$. Further, we can obtain the following conservative bounds for the unknown time-varying signals for the implementation of the projection operator, $\omega_i \in [0.8\ 1.2]$, $i = [1, \ldots, 6]$; $|\theta_i(t)| \leq 3800$, $i = [1, \ldots, 6]$.

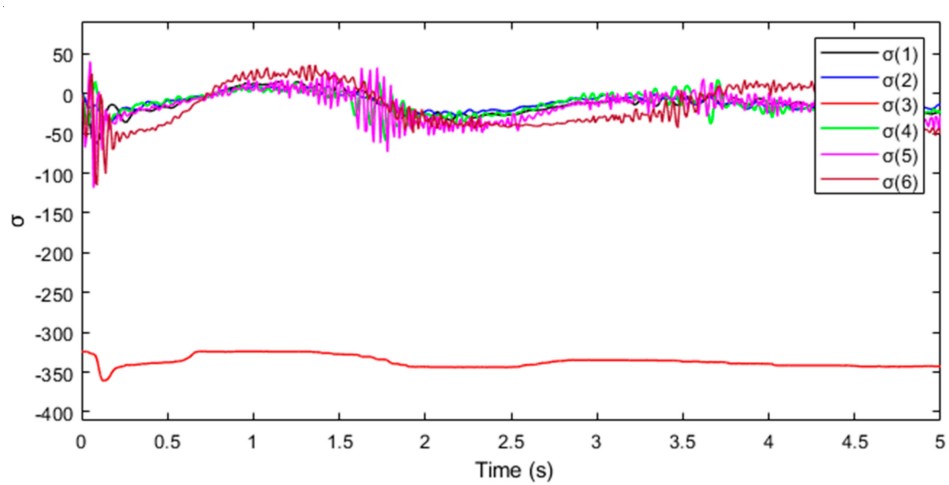

**Figure 6.** Curve trajectory of parameter σ in complex state.

### 4.4. Simulation Results

In the experimental verification, we tested the step response and frequency response of the control system. The numerical simulation included two parts. The first part tested the MRAC and $\mathcal{L}_1$ control performance of $r = [0.2\ 0.3\ 0.78\ 0.25\ 0.5\ 1.0]$, as illustrated in Figure 7. The second part tests tested the MRAC and $\mathcal{L}_1$ control performance of $r = \left[0.2sin(\pi t),\ 0.3sin(\pi t),\ 0.48 + 0.14sin(\pi t),\ \frac{\pi}{12}sin(\pi t),\ \frac{\pi}{6}sin(\pi t),\ \frac{\pi}{3}sin(\pi t)\right]$, as illustrated in Figure 8. Because of space limitation, we only give the simulation data of position DOFs, and the attitude DOFs are similar to it.

The simulation results of the MRAC are shown in Figure 7a,c,e and the $\mathcal{L}_1$ adaptive controller shown in Figure 7b,d,f for step reference inputs $r = [0.2\ 0.3\ 0.78\ \pi/12\ \pi/6\ \pi/3]$. We found that both of MRAC and the $\mathcal{L}_1$ adaptive controller had the same instantaneous response characteristics. Therefore, we could improve the transient response time by changing the state matrix $A_m$ to meet the requirements of abnormal attitude, large angular velocity, and acceleration changes of the aircraft in the abnormal flight state.

From Figure 7g,h, it can be seen that the instantaneous output peak of $\mathcal{L}_1$ adaptive controller was significantly lower than that of MRAC. Taking the output of No.2 leg driver (u2) of the Stewart simulator as an example, the instantaneous output peak value of MRAC controller was 438 mNm, while the instantaneous output peak value of leg 2 corresponding to $\mathcal{L}_1$ adaptive controller was 263 mNm, which reduced 39.95% of the peak output. The results show that the $\mathcal{L}_1$ adaptive controller improved the instantaneous performance of the actuator and reduced the requirement for the performance of the actuator without changing the response speed.

It can be seen from Figure 8 that the $\mathcal{L}_1$ adaptive controller can filter out the high-frequency signals in the control variables and add the low-frequency signals to enhance the smoothness of flight simulator motion and improve the motion sense effect of abnormal state recovery training on the flight simulator.

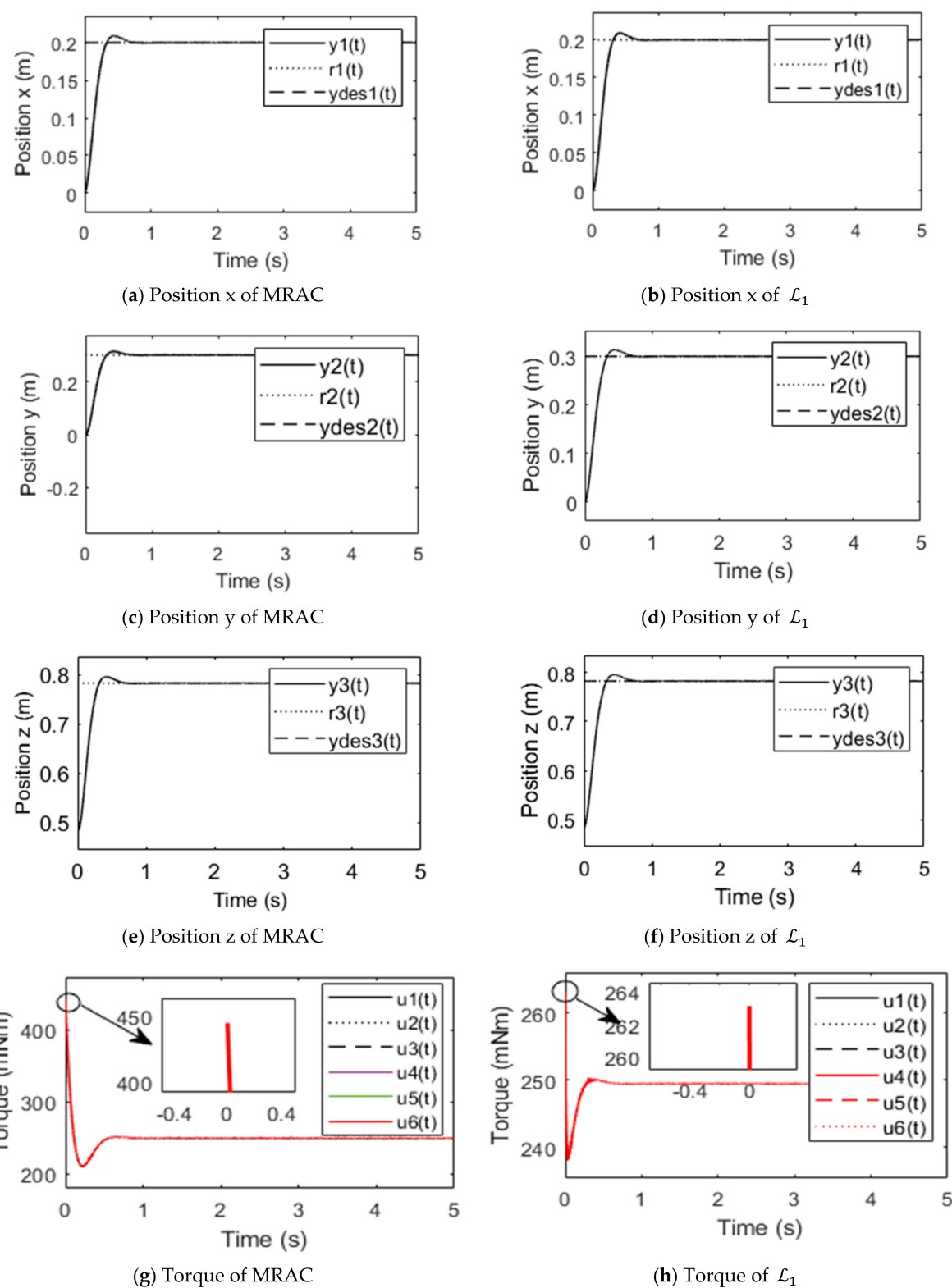

**Figure 7.** Performance of MRAC and $\mathcal{L}_1$ adaptive controller for $r = [0.2\ 0.3\ 0.78\ 0.25\ 0.5\ 1.0]$. (**a**) X-position curve of the platform when using MRAC controller; (**b**) X-position curve of the platform when using $\mathcal{L}_1$ adaptive controller; (**c**) Y-position curve of the platform when using MRAC controller; (**d**) Y-position curve of the platform when using $\mathcal{L}_1$ adaptive controller; (**e**) Z-position curve of the platform when using MRAC controller; (**f**) Z-position curve of the platform when using $\mathcal{L}_1$ adaptive controller; (**g**) Torque **u** curve of the platform when using MRAC controller; (**h**) Torque **u** curve of the platform when using $\mathcal{L}_1$ adaptive controller.

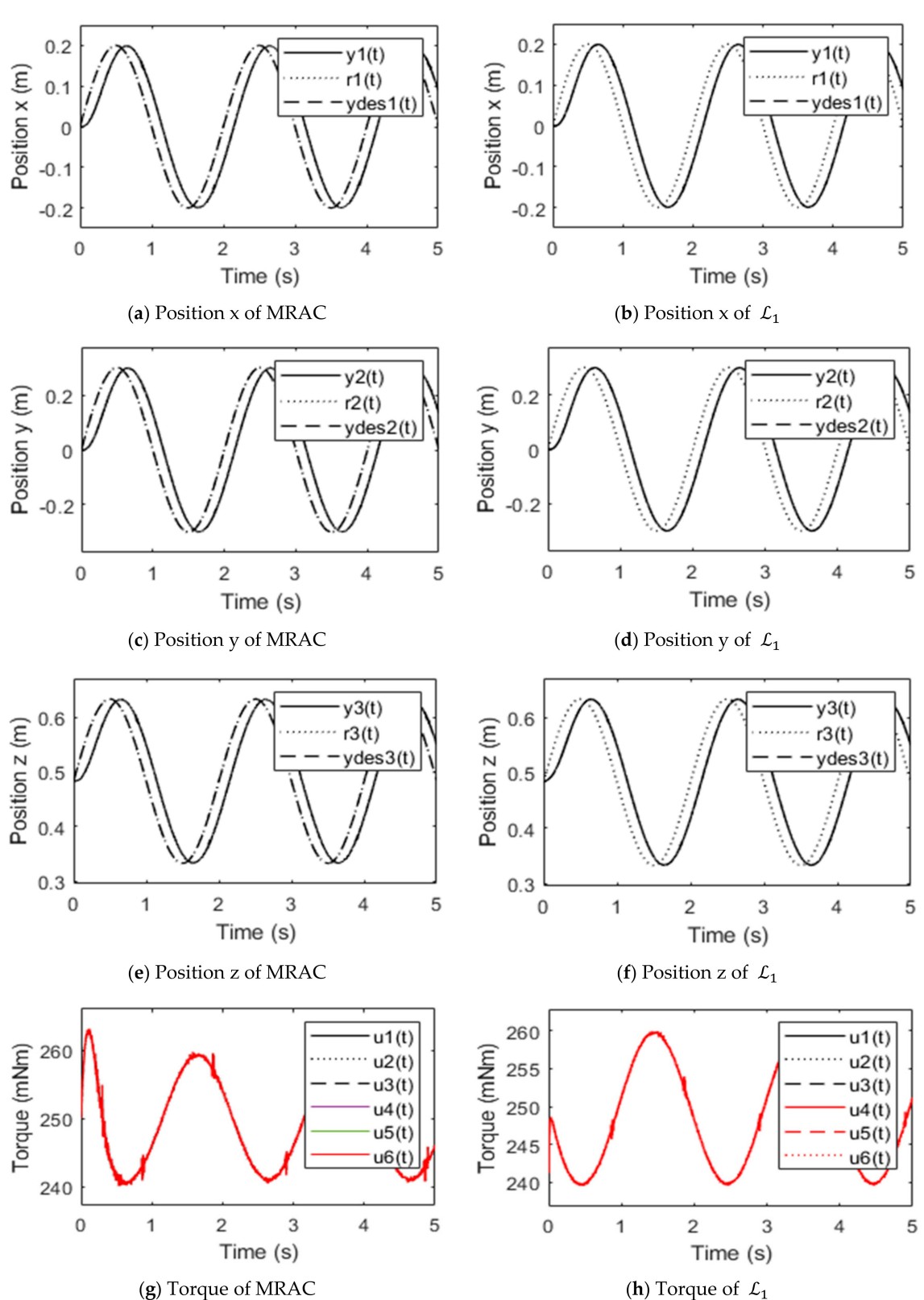

**Figure 8.** Performance of MRAC and $\mathcal{L}_1$ adaptive controller for $r = [0.2sin(\pi t), 0.3sin(\pi t), 0.48 + 0.14sin(\pi t), \frac{\pi}{12}sin(\pi t), \frac{\pi}{6}sin(\pi t), \frac{\pi}{3}sin(\pi t)]$. (**a**) X-position curve of the platform when using MRAC controller; (**b**) X-position curve of the platform when using $\mathcal{L}_1$ adaptive controller; (**c**) Y-position curve of the platform when using MRAC controller; (**d**) Y-position curve of the platform when using $\mathcal{L}_1$ adaptive controller; (**e**) Z-position curve of the platform when using MRAC controller; (**f**) Z-position curve of the platform when using $\mathcal{L}_1$ adaptive controller; (**g**) Torque **u** curve of the platform when using MRAC controller; (**h**) The torque **u** curve of the platform with $\mathcal{L}_1$ adaptive controller.

## 5. Conclusions

In this paper, an $\mathcal{L}_1$ adaptive controller was used in the control of the 6-DOF Stewart-Gough flight simulation platform. The main contribution of this work is the design of a 6-DOF simulation platform for $\mathcal{L}_1$ adaptive controller. Compared with the traditional MRAC architecture, the $\mathcal{L}_1$ adaptive controller achieves the separation design of tracking error signal and control input signal. Based on the stability design of tracking error of traditional adaptive control, the $\mathcal{L}_1$ adaptive control considers the stability design of control input, which tolerates higher adaptive gain and improves the transient performance of the system. When designing the controller, this architecture will not affect the error signal. It allows high adaptive gain and can easily track the control input so as to improve the transient performance and robustness of the system.

The addition of low-pass filter not only improves the instantaneous performance of $\mathcal{L}_1$ adaptive controller without changing the response speed but also reduces the requirement for the performance of the actuator. It can also filter out the high-frequency signal in the control variables and add the low-frequency signal to the actual system to improve the smoothness of flight simulator motion and improve the training effect of abnormal state recovery on the flight simulator. Finally, the algorithm complexity of $\mathcal{L}_1$ adaptive controller is a little high, and reducing the amount of on-line calculation is the main research direction of this paper.

**Author Contributions:** Conceptualization, J.Z., D.W. and H.G.; methodology, J.Z., D.W. and H.G.; software, J.Z.; validation, J.Z., D.W. and H.G.; formal analysis, J.Z.; investigation, J.Z., D.W. and H.G.; resources, J.Z.; data curation, J.Z.; writing—original draft preparation, J.Z.; writing—review and editing, J.Z.; visualization, J.Z., D.W. and H.G.; supervision, H.G.; project administration, D.W.; funding acquisition, D.W. All authors have read and agreed to the published version of the manuscript.

**Funding:** This research was funded by the National Natural Science Fund Civil Aviation Joint Fund Project of China, grant number U1633120; U163310099; and the Special Funds for Basic Scientific Research Operations of Central Universities (NP2018409).

**Institutional Review Board Statement:** "Not applicable" for studies not involving humans or animals.

**Informed Consent Statement:** "Not applicable" for studies not involving humans.

**Data Availability Statement:** The data presented in this study are available on request from the corresponding author. The data are not publicly available due to there is no data uploaded on the website.

**Conflicts of Interest:** The authors declare no conflict of interest. The funders had no role in the design of the study; in the collection, analyses, or interpretation of data; in the writing of the manuscript, or in the decision to publish the results.

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
