# Peer review of "Performance Evaluation of Stewart-Gough Flight Simulator Based on L1 Adaptive Control"

_applsci, doi:10.3390/app11073288_

Round 1

Reviewer 1 Report

The paper is well written. The introduction covers the need for the adaptive controller described to be implemented on flight simulators. The literature review provides and ample overview of similar control methods for similar applications. The methodology initially provides the necessary background for the proposed approach, then presents the flight simulator model and the derived controller. Experimental results are then presented to validate the proposed controller design. The content is clear and every step is easy to follow. All terms involved in the equations are explicitly defined and every step in the methodology is justified. Below are a few comments of items that stand out.

  • The introduction leads the reader to believe that “abnormal conditions” causing “LOSS-OF-CONTROL In-Flight” must be simulated on a flight simulator to train pilots, thus the need for an adaptive controller that can accommodate these conditions in the flight-simulator. However, the trajectories (step function and time dependent sinusoidal) used for testing the controller performance, presented in section 4.4, do not seem to be representative of such conditions.
  • The wording “control low” in figure 1, is confusing. Is it Low-level controller? Is it control law?
  • Based on the literature review, the proposed approach seem to be equivalent to that presented in [26]? In such case, the controller presented in this paper lacks scientific novelty. If not, how is it different from [26]?
  • Although, based on the wording it seems like the controller was implemented and tested on the flight simulator itself, the results presented in figure 7 and 8 seem to be simulated. If this is the case, it should made explicit that the controller performance was verified using (MATLAB or other?). In such case, the word “experimental results” in the title is misleading and it should be changed to “simulation results”, as experimental results have a different set of value as compared to simulations based on system dynamic models. If the controller was actually implemented in the flight simulator, additional details about the implementation should be provided.
  • The author claims that the peak output of the actuators is reduced by ~40%, thus reducing their performance requirement, based on the controller step response shown in Fig.7, however that claim is no longer verified in fig. 8 when an actual trajectory is fed to the controller.

Author Response

Thanks to the reviewer for your valuable opinions, which are very helpful to us. After receiving the reviewers' opinions, we did the experiment again, improved the designed system, and made corresponding modifications in the paper. The following is a detailed description of our changes.

Q:“The introduction leads the reader to believe that “abnormal conditions” causing “LOSS-OF-CONTROL In-Flight” must be simulated on a flight simulator to train pilots, thus the need for an adaptive controller that can accommodate these conditions in the flight-simulator. However, the trajectories (step function and time dependent sinusoidal) used for testing the controller performance, presented in section 4.4, do not seem to be representative of such conditions.”

Revise: Because of our reasons, it is easy to cause misunderstanding if this part is not described clearly. It has been modified in lines 29-31 of the article and highlighted in red font.

“The abnormal states such as high altitude stall, wind shear and turbulence are usually accompanied by the abnormal attitude of the aircraft and the large changes of angular velocity and acceleration, which requires the higher requirements of the transient characteristics and anti-high frequency interference ability of the flight simulator control system.”

The traditional flight simulation platform is based on MRAC architecture, which can be used for abnormal state out training. However, when the aircraft is in stall, wind shear, turbulence and other abnormal states, the traditional MRAC control scheme often makes the moving platform produce high frequency jitter, which will produce excessive output gain of the controller in transient state. The proposed L1 adaptive control reduces the high frequency chattering due to the addition of low-pass filter, which has been verified in the simulation of step function and time-varying sinusoidal trajectory. The key point of this paper is to improve and verify the performance of the controller. The controlled factor of the quality of flight simulation training mainly comes from the washout algorithm, and its evaluation standard is greatly affected by the subjective factors of pilots. In view of the fact that the platform is a second-order system, the step function and time-varying sine curve are used as the main experimental verification scheme.

Q: The wording “control low” in figure 1, is confusing. Is it Low-level controller? Is it control law?

Revise: Figure 1 has been modified, it is control law

Thank you very much for the conscientiousness and responsibility of the reviewers. I also apologize for our low-level mistakes

Q: Based on the literature review, the proposed approach seem to be equivalent to that presented in [26]? In such case, the controller presented in this paper lacks scientific novelty. If not, how is it different from [26]?

Revise: In the article 50-54 line added new content, has carried on the detailed explanation, and has carried on the prominent display with the red font.

“and the  adaptive controller is applied to the control field of flight simulation platform. Due to the advantage of the separation of control law and adaptive law, the low-pass filter is added to the control input to avoid the high-frequency noise in the control signal. and the structure diagram of  adaptive flight simulation control system is shown in Figure 1.”

The innovation of this paper is: we apply  adaptive controller to the control field of flight simulation platform for the first time. Due to the advantage of the separation of control law and adaptive law, we add a low-pass filter in the control input to avoid the high-frequency noise in the control signal. The  controller keeps the good transient response of MRAC and reduces the peak output of actuator, which effectively reduces the performance requirements of actuators.

Q: Although, based on the wording it seems like the controller was implemented and tested on the flight simulator itself, the results presented in figure 7 and 8 seem to be simulated. If this is the case, it should made explicit that the controller performance was verified using (MATLAB or other?). In such case, the word “experimental results” in the title is misleading and it should be changed to “simulation results”, as experimental results have a different set of value as compared to simulations based on system dynamic models. If the controller was actually implemented in the flight simulator, additional details about the implementation should be provided.

Revise: The experimental results are obtained by simulation on MATLAB, the word “experimental results” in the title is be changed to “simulation results”. In line 203-205 added new content.

“The modeling parameters required for the experiment are from the 6-DOF Stewart Gough flight simulator platform shown in Figure 2, and the dynamic model and the  adaptive controller were implemented in MatLab.”

Q: The author claims that the peak output of the actuators is reduced by ~40%, thus reducing their performance requirement, based on the controller step response shown in Fig.7, however that claim is no longer verified in fig. 8 when an actual trajectory is fed to the controller.

Revise: Because the transient characteristics of the controller will reduce the peak output of the actuator and improve the control performance in the signal with large change rate, the frequency of the time-varying sinusoidal signal used in our experiment is relatively low, which is not obvious. In our usual experiments, we can see that under the action of high frequency signal, the peak output of L1 adaptive controller is significantly improved compared with that of MRAC controller.

Reviewer 2 Report

The reviewer feels that the paper is interesting, and it is within the scope of the Journal, but one additional revision should be done. the main issue is related to the quality of English and, therefore, a careful revision should be done. Please see the following comments:

Specific comments:

Please improve the specific statements:

Line 27: “Hence, it is necessary to train pilots proper application recovery from upsets”

Lines 30-32: “Because the abnormal state in flight is usually accompanied by the abnormal attitude, large angular velocity and acceleration changes of the aircraft, it puts forward higher requirements for the flight simulator system”

Lines: 43-45: “However, in the transient state, the system is prone to high frequency oscillation of control input, when the adaptive rate is small and the adaptive gain is increased, the unmodeled dynamics of the model are easily caused.”

Please remove the word And in line 57.

In line 65, reference 26 is not appropriated, please, change to reference 27.

In line 68, the use of the symbol delta (δ) on force and torque vector should not be used. The reviewer knows that in reference 27 it appears according it is presented in this paper, but is not correct either. In the principle of virtual works, the forces are real and displacements are the virtual variables. Hence, the delta symbol is used to denote virtual quantities.

In line 69: it would be preferable to say that are virtual variables or virtual state variables.

Please improve the English of the statement between lines 71-74.

In line 94, place the name of authors of reference 28: according to …

In all parts of the document, authors forget to place the auxiliary verb “is” to say for instance: is given, etc. please correct this kind of error. Moreover, Figure 4 appears previously to be referenced.

What are the main limitations of this work? This information should be placed in the paper.

Author Response

Thanks to the reviewer for the valuable opinions, which are very helpful to us. After receiving the reviewers' opinions, we did the experiment again, improved the designed system, and made corresponding modifications in the paper. The following is a detailed description of our changes.

Q:Please improve the specific statements:

Revise: We have improved the specific expression of the corresponding line of the article and highlighted it with a yellow background. The specific modifications are as follows:

Line 27: “Hence, it is necessary to train pilots proper application recovery from upsets”

Revise: Line 27-28:”Therefore, the pilot should adequate training by practicing the recovery technique many times on flight simulators before maneuvers on a real aircraft.”

Lines 30-32: “Because the abnormal state in flight is usually accompanied by the abnormal attitude, large angular velocity and acceleration changes of the aircraft, it puts forward higher requirements for the flight simulator system”

Revise: Line 32-33:” and turbulence are usually accompanied by the abnormal attitude of the aircraft and the large changes of angular velocity and acceleration,”

Lines: 43-45: “However, in the transient state, the system is prone to high frequency oscillation of control input, when the adaptive rate is small and the adaptive gain is increased, the unmodeled dynamics of the model are easily caused.”

Revise: Line 45-47:” However, when the adaptive rate is small, a large amplitude of control signal is easy to appear during the transient; when the adaptive law is large, the system will have high frequency oscillation.”

Please remove the word And in line 57.

Revise: And has been deleted

In line 65, reference 26 is not appropriated, please, change to reference 27.

Revise: [26] has been changed to [27]

In line 68, the use of the symbol delta (δ) on force and torque vector should not be used. The reviewer knows that in reference 27 it appears according it is presented in this paper, but is not correct either. In the principle of virtual works, the forces are real and displacements are the virtual variables. Hence, the delta symbol is used to denote virtual quantities.

Revise: Line 73:””

In line 69: it would be preferable to say that are virtual variables or virtual state variables.

Revise: Line 74:” virtual displacement has been changed to virtual variables ”

Please improve the English of the statement between lines 71-74.

Revise: Line 75-80:” Furthermore,  represents the virtual variables vector of the upper legs center of gravity(c.o.g.); similarly,  denotes the virtual displacement vector c.o.g. of down legs;  denotes the inertial forces at the c.o.g. of the moving platform; and and  are the inertial forces at the c.o.g. of up legs and at the c.o.g. of the down legs, respectively, which are defined from the relationship according[27] is”

In line 94, place the name of authors of reference 28: according to …

Revise: In line 99: ” according to [28] has been changed to according to Tsai LW  in [28]”

In all parts of the document, authors forget to place the auxiliary verb “is” to say for instance: is given, etc. please correct this kind of error.

Revise: It has been added

What are the main limitations of this work? This information should be placed in the paper.

Revise: in line 344-346 we added “Finally, the algorithm complexity of  adaptive controller is a little high, and reducing the amount of on-line calculation is the main research direction of this paper.”

Reviewer 3 Report

Make sure that all your abbreviations are explained ( i.e. DOF). Also please correct English ( i.e. row 17 - to maintaining, row 19, row 33, row 57 - sentences that starts with And, the word denotes used too often in a sentence - rows 154-159, correct also row 318), The conclusions are briefly presented and shows very clearly the results of your important research.

Author Response

Thanks to the reviewer for the valuable opinions, which are very helpful to us. After receiving the reviewers' opinions, we did the experiment again, improved the designed system, and made corresponding modifications in the paper. The following is a detailed description of our changes.

Q:Make sure that all your abbreviations are explained ( i.e. DOF). Also please correct English ( i.e. row 17 - to maintaining, row 19, row 33, row 57 - sentences that starts with And, the word denotes used too often in a sentence - rows 154-159, correct also row 318), The conclusions are briefly presented and shows very clearly the results of your important research.

Revise: We have improved the specific expression of the corresponding line of the article, and highlighted it with green font. The specific modifications are as follows:

Make sure that all your abbreviations are explained ( i.e. DOF).

Revise: In line 35: “added 6-DOF(degree of freedom)”

Also please correct English ( i.e. row 17 - to maintaining, row 19, row 33, row 57 - sentences that starts with And, the word denotes used too often in a sentence - rows 154-159, correct also row 318)

Revise: In line 33: “and turbulence are usually accompanied by the abnormal attitude of the aircraft and the large changes of angular velocity and acceleration,”

Revise: In line 61-62: “and the filter can also be used to reduce the output amplitude of the actuator in transient state”

Revise: In line 159-164: “In which,  represents the state of the system,  represents the control input,  represents the output matrix,  represents the known constant matrix,  denotes the regulated output,  denotes the input uncertainly parameter of the model, is the uncertainly parameter of the model itself, and  represents the unmatched disturbances,  represents a Hurwitz matrix,  is controllable and .”

Revise: In line 324-325: “It can be seen from Figure 8 that  adaptive controller can filter out the high-frequency signals in the control variables”
